# *Jatropha curcas* L. as a Plant Model for Studies on Vegetative Propagation of Native Forest Plants

**DOI:** 10.3390/plants11192457

**Published:** 2022-09-21

**Authors:** Renato Lustosa Sobrinho, Tiago Zoz, Taciane Finato, Carlos Eduardo da Silva Oliveira, Sebastião Soares de Oliveira Neto, André Zoz, Ibrahim A. Alaraidh, Mohammad K. Okla, Yasmeen A. Alwasel, Gerrit Beemster, Hamada AbdElgawad

**Affiliations:** 1Programa de Pós-Graduação em Agronomia, Universidade Tecnológica Federal do Paraná, Pato Branco 85503-390, PR, Brazil; 2Department of Environmental Management, Mato Grosso do Sul State University, Mundo Novo 79980-000, MS, Brazil; 3Department of Plant Protection, Rural Engineering and Soils, School of Engineering, São Paulo State University—UNESP-FEIS, Ilha Solteira 15385-000, SP, Brazil; 4Agricultural Defense Coordination of São Paulo, Presidente Prudente 19013-050, SP, Brazil; 5Department of Plant Production, School of Agricultural, São Paulo State University—FCA/UNESP, Botucatu 18610-034, SP, Brazil; 6Botany and Microbiology Department, College of Science, King Saud University, Riyadh 11451, Saudi Arabia; 7Integrated Molecular Plant Physiology Research, Department of Biology, University of Antwerp, 2020 Antwerp, Belgium; 8Botany and Microbiology Department, Faculty of Science, Beni-Suef University, Beni-Suef 62511, Egypt

**Keywords:** vegetative propagation, clone, plant regulators, auxin, gibberellin

## Abstract

**Highlights:**

*Jatropha curcas* L. has great potential to be used as a model plant in several studies involving native forest species.The immersion in the 2,4-D solution accelerated the emission of primary roots in hardwood cuttings.Studies on vegetative propagation of native species can use *Jatropha curcas* L. species as a model for obtaining important information in a short time and reducing labor costs.The immersion of cuttings of native species in solutions with low concentrations of 2,4-D can favor the rooting process and vegetative propagation.

**Abstract:**

Even though it is a forest native plant, there are already several studies evaluating the small genome of *Jatropha curcas* L., which belongs to the Euphorbiaceae family, and may be an excellent representative model for the other plants from the same family. *Jatropha curcas* L. plant has fast growth, precocity, and great adaptability, facilitating silvicultural studies, allowing important information to be obtained quickly, and reducing labor costs. This information justifies the use of the species as a model plant in studies involving the reproduction of native plants. This study aimed to evaluate the possibility of using *Jatropha curcas* L. as a model plant for studies involving native forest plants and establish possible recommendations for the vegetative propagation of the species using hardwood cuttings. The information collected can be helpful to other native forest plant species, similar to *Jatropha curcas* L. To this end, the effects of hardwood cutting length (10, 20, and 30 cm) and the part of the hardwood cuttings (basal, middle, and apex) were evaluated. Moreover, the influence of immersing the hardwood cuttings in solutions containing micronutrients (boron or zinc) or plant regulators (2,4-D, GA3) and a biostimulant composed of kinetin (0.09 g L^−1^), gibberellic acid (0.05 g L^−1^), and 4-indole-3-butyric acid (0.05 g L^−1^). The experiments were carried out in duplicates. In one duplicate, sand was used as the substrate, and rooting evaluations were made 77 days after planting. In another duplicate, a substrate composed of 50% soil, 40% poultry litter, and 10% sand was used, and the evaluations of the saplings were performed 120 days after planting. The GA3 solutions inhibited the roots’ and sprouts’ emissions, while immersion in 2,4-D solution increased the number of primary roots at 77 days after planting. The hardwood cuttings from the basal part of the branch had the best results for producing saplings.

## 1. Introduction

There are indications that the *Jatropha curcas* L. plant originated in South and Central America; the name of the species *Jatropha curcas* L. comes from the Greek language “iatrós”, which means (doctor) and (trophé), which means food; there are reports that the Portuguese were using the medicinal properties of the *Jatropha* plant since the 16th century [1]. The Portuguese also established the first commercial plantations of the species to produce soap and lamp oil in Cape Verde and Guinea-Bissau; from there, the species spread to Africa, South America, and Asia, probably for its medicinal properties [2]. 

The *Jatropha curcas* L. plant is an oilseed that does not compete directly with food crops; its grains have around 34% oil content [1], which can produce biodiesel with a high calorific value, such as diesel [3], its oil can be added to blends with kerosene to obtain high-quality aviation fuel [4], or with C-heavy oil to be used in oil-fired boilers, with the advantage of reducing emissions of NOx and SO_2_ [5]. In addition, *Jatropha curcas* L. is being used to recover degraded forest areas [6], and in the pharmaceutical industry due to its pharmacological properties [7].

The *Jatropha* plant is found in the Brazilian landscape and has withstood the great damage caused by illegal deforestation; thus, the *Jatropha* plant can be an alternative for facilitating the recovery of areas of degraded native forest, aiming to restore the ecological balance of existing natural ecosystems [8]. In this context, the cuttings and mini-cuttings widely used in commercial nurseries in the eucalyptus industry may prove to be a promising alternative for the reproduction of *Jatropha* for reforestation, and also for the rescue of other endangered native forest species [8,9].

However, there are several factors related to the reproductive characteristics of the native plants that serve as obstacles to their wide reproduction under controlled conditions, such as the uneven development of the plants obtained via seminiferous, increasing costs related to labor during management; in addition to the difficulties in defining the ideal point of seed and fruit maturation, to obtain a greater germination potential [10,11]. In addition, most of the native forest species correspond to plants that have not undergone a domestication process [12]. Associated with this, there is a lack of technical information about native species, especially because the scientific studies are usually carried out species by species, which makes the process of obtaining new information slow and time-consuming [13], making their large-scale reproduction in commercial nurseries difficult.

To speed up the process of obtaining information about native species in general, in this study, we propose the use of the concept of a model plant for each botanical family. It is known that a “model organism” for scientific research needs to have the characteristics that facilitate the intensive and extensive study of biological phenomena, which in theory would provide a better understanding of these same phenomena in other similar organisms; as in the case of the well-known plant model, *Arabidopsis thaliana* [14].

In this sense, *Jatropha* plants have great potential to be used as a model plant in several studies involving native forest species.

### 1.1. Why Can Jatropha Be an Excellent Plant Model for Studies on Vegetative Propagation of Native Forest Plants?

Belonging to the Euphorbiaceae family, the *Jatropha curcas* L. plant can be an excellent representative model for this broad family of plants, whose occurrence is common in the ecosystems of the American continent, especially in Brazil, where we can mention the well-known rubber tree “*Hevea* sp.”, belonging to the same family [15,16].

Although it is a native plant, several studies are already evaluating its small genome [17,18,19]. With a bushy size and fast growth, being able to reach up to 20 cm in trunk diameter and 5 m in height in three years in favorable conditions, the *Jatropha* begins to produce seeds in just eight months after sowing; this period is even shorter when the plant is obtained via vegetative propagation, in addition, the plant can live in the field for up to 40 years [20].

This fast growth, precocity, and adaptability can facilitate silvicultural studies, allowing for the obtaining of important information in a short time, and reducing labor costs. This information justifies the use of the species as a model plant in studies involving the reproduction of native plants.

### 1.2. Plant Regulators and Vegetative Propagation of Jatropha Plants

The use of cuttings for plant propagation is the preferred method of several producers, mainly due to their greater simplicity and economy [21,22].

In this context, some of the factors can influence the vegetative propagation of plants; mainly those related to plant metabolism at the time of emission of new roots, such as the type and size of cuttings and the carbohydrate reserves in the cutting [23]; levels of plant growth-regulating compounds; and levels of nutrients, especially boron and zinc, which play an essential role in activating the cell differentiation process to produce new roots in several plant species [24].

In turn, the plant hormones, especially the auxins and their directional transport, play a regulatory role in mediating the various processes of cell division and expansion [25]; due to the directional nature of these processes, the choice of an adequate part of the branch to obtain the hardwood cuttings and its length to produce saplings by vegetative propagation are essential and still not known yet for the *Jatropha curcas* L. or for other similar species [6,8]. The substances capable of promoting or inhibiting the rooting process vary along the branches of the plant, and these variations are responsible for the different rooting rates of branches from different parts of the plant. In general, the hardwood cuttings are more lignified, making the rooting process difficult, due to ontogenetic aging and consequent reduction in the biological activity of these tissues [26].

Based on these data, this study aimed to evaluate the possibility of using *Jatropha* as a model plant for studies involving native forest plants and establishing possible recommendations for the vegetative propagation of the species using hardwood cuttings. The information collected can be useful to other native forest plant species, similar to *Jatropha curcas* L.

## 2. Material and Methods

### 2.1. Experimental Conditions

Two experiments were carried out under an agricultural screenhouse with Sombrite^®^ type mesh, 50% shading, between August and December 2016. The experiment I was conducted to evaluate the adequate hardwood cutting length to produce *Jatropha curcas* L. saplings. The experiment II was conducted to assess the effect of micronutrients and plant regulators on the rooting and formation of *Jatropha curcas* L. saplings. Both of the experiments were conducted in duplicate. A duplicate was used to evaluate the rooting at 77 days after planting. Another duplicate was used to assess the production of saplings at 120 days after planting. Both of the experiments were watered daily by aspersion until they reached 100% field capacity based on tensiometer data.

According to the climatic classification of Köppen–Geiger, the region presents a tropical climate with a dry winter season (Aw-type). The temperature and relative air humidity were monitored daily during the whole period of the experiment using an ITLOG-80 Data Logger installed inside the agricultural greenhouse (Figure 1).

### 2.2. Experimental Design

#### 2.2.1. Experiment I—Hardwood Cutting Length

The randomized complete block design was used, with four replications arranged in a 3 × 3 factorial scheme. The stem cuttings with 10, 20, and 30 cm extracted from the basal, middle, and apex of branches from the upper third of the plant were collected. The factors are described below:

Basal cuttings + 10 cm;

Basal cuttings + 20 cm;

Basal cuttings + 30 cm;

Middle cuttings + 10 cm;

Middle cuttings + 20 cm;

Middle cuttings + 30 cm;

Apex cuttings + 10 cm;

Apex cuttings + 20 cm;

Apex cuttings + 30 cm.

Each experimental unit (replication) was composed of three plastic bags containing one hardwood cutting in each bag.

Branches from the five-year-old Jatropha plants were collected. The collection was carried out in the morning on 6 August 2016. From each plant, between two and four branches were obtained, approximately one meter long from the upper third of the plant. The cuts were performed horizontally with pruning shears. The five centimeters of the basal end of each branch were discarded, and the remainder was divided into three parts: basal, middle, and apex, of 10, 20, and 30 cm in length. The diameter of the intermediate third of the stem cuttings was measured (Figure 2).

#### 2.2.2. Experiment II—Micronutrients and Plant Regulators

The experimental design was a randomized block design arranged in a 6 × 3 factorial scheme with four replications. The factors are described below:

4 h immersion in Boric acid + Basal cuttings (15 cm);

4 h immersion in Boric acid + Midle cuttings (15 cm);

4 h immersion in Boric acid + Apex cuttings (15 cm);

4 h immersion in Zinc sulfate + Basal cuttings (15 cm);

4 h immersion in Zinc sulfate + Midle cuttings (15 cm);

4 h immersion in Zinc sulfate + Apex cuttings (15 cm);

4 h immersion in 2,4-D + Basal cuttings (15 cm);

4 h immersion in 2,4-D + Midle cuttings (15 cm);

4 h immersion in 2,4-D + Apex cuttings (15 cm);

4 h immersion in Stimulate^®^ + Basal cuttings (15 cm);

4 h immersion in Stimulate^®^ + Midle cuttings (15 cm);

4 h immersion in Stimulate^®^ + Apex cuttings (15 cm);

4 h immersion in Ga3 + Basal cuttings (15 cm);

4 h immersion in Ga3 + Midle cuttings (15 cm);

4 h immersion in Ga3 + Apex cuttings (15 cm);

4 h immersion in Control (Water) + Basal cuttings (15 cm);

4 h immersion in Control (Water) + Midle cuttings (15 cm);

4 h immersion in Control (Water) + Apex cuttings (15 cm).

The first factor was composed of the immersion, for four hours, of the hardwood cuttings in distilled water or solutions containing micronutrients or plant regulators. The second factor was the hardwood cuttings obtained from different branch parts (basal, intermediate, and apex). Each experimental unit (replication) was composed of five plastic bags containing one hardwood cutting in each bag.

Two micronutrient solutions and three plant regulator solutions were evaluated in isolation. For the micronutrient solutions, boric acid (23.3% boron) at a concentration of 200 mg L^−1^ of boron and zinc sulfate (22.7% of zinc) at a concentration of 100 mg L^−1^ of zinc were used. The following plant regulators were evaluated in the solutions: 2,4-D herbicide containing 10.5 mg a.i. L^−1^ (DMA^®^806 BR with 806 g L^−1^ of dimethylamine salt); a biostimulant (Stimulate^®^), composed of the phytohormones kinetin (0.09 g L^−1^), gibberellic acid (0.05g L^−1^), and 4-Indole-3-butyric acid (0.05 g L^−1^), at the concentration of 4 mL L^−1^; and GA3 containing 2.0 g a.i. L^−1^ of gibberellic acid (Pro-Gibb^®^ with 100 g kg^−1^ of gibberellic acid). Commercial products that contain plant regulators were chosen because of the ease with which producers can find them in the local market. The concentrations of micronutrients and plant regulators were determined based on work with species related to *Jatropha curcas* L.

The branches were collected on 6 August 2016 in the morning, from five-year-old plants. From each plant, between two and four branches approximately one meter long were obtained from the upper third of the plant. The cuts were performed horizontally with pruning shears. The five centimeters of the basal end of each branch were discarded, and the remainder was divided into three parts: basal, middle, and apex. From these parts, the hardwood cuttings of 15 cm in length were obtained. The cuttings were placed upright with half of their length immersed in the solutions with micronutrients or plant regulators. Care was taken so that each hardwood cutting had its bottom part immersed in the solutions.

### 2.3. Implementation and Conduction of Experiments

In both of the experiments, the bottom of the stem cuttings was buried 5 cm deep in the substrate, upright, in plastic bags with 2.35 dm^3^. In half of the plots of each experiment, sand was used as a substrate, aiming to study the rooting. In the other half of each experiment, a substrate composed of 50% soil, 40% poultry litter, and 10% sand was used to study the growth of the saplings. Irrigation was performed using a suspended micro-sprinkler system with Netafim SpinNet emitters with an irrigation capacity of 70 L per hour, programmed to irrigate every 24 h.

At 77 days after planting, the percentages of rooted cuttings, sprouted cuttings, callused cuttings, and the survival rate of cuttings were analyzed in the experiment for rooting evaluation. In addition, the number of sprouts, leaves, and primary roots per stem cutting and the dry mass of shoots, roots, and the total were evaluated. In the experiment of the saplings’ production, at 120 days after planting, the number of sprouts, leaves, and primary roots per stem cutting and the dry mass of shoots, roots, and the total were evaluated.

### 2.4. Statistical Analysis

To meet the assumptions of normality and homoscedasticity, the data were transformed into $\sqrt{x+0.5}$ and then submitted to the analysis of variance. The F-test tested the significance of the mean squares obtained in the analysis of variance at the 5% probability level. The means were compared by the Least Significant Difference (LSD) test at the 5% probability level, using SISVAR software [27].

## 3. Results

### 3.1. Hardwood Cutting Length

Hardwood cuttings of 20 and 30 cm lengths had a survival rate of around 86% and 89%, respectively, and these were superior to the cuttings of 10 cm in length (61%) (Figure 3).

The larger hardwood cuttings (30 cm) had a higher percentage of rooting than the smaller cuttings (10 cm) (Figure 3). Around 83% and 81% of the sprouted cuttings were observed for the 20 and 30 cm cuttings, respectively. The smaller cuttings (10 cm) had a lower percentage of sprouted cuttings than the cuttings of 20 and 30 cm in length. There was no influence of cutting length on the percentage of callused cuttings (Figure 4). Overall, no interaction was observed between the cutting length and part of the branch for the variables at 77 days after the planting.

The difference between the cuttings of 20 and 30 cm in length was not observed for the number of sprouts and leaves per cutting. The cuttings of 20 cm in length had about 59% and 66% more sprouts than the 10 cm cuttings, respectively. Around 60% and 70% more leaves were also observed on the cuttings of 20 cm in length than on the cuttings of 10 cm in length. The cuttings of a 30 cm length had a number of primary roots around 42% higher than the 10 cm cuttings (Figure 3).

For the dry mass of the roots, shoots, and the total, there was no difference between the hardwood cuttings of 20 and 30 cm in length, and both were, on average, 433%, 322%, and 333% higher than the cuttings of 10 cm in length, respectively (Figure 3). At 120 days after the planting, there was no interaction between the hardwood cutting length and the part of the branch. The hardwood cuttings of 30 cm in length had a number of sprouts about 178% and 33% higher than the cuttings of 10 and 20 cm in length, respectively (Figure 3). The number of sprouts of the 20 cm cuttings was around 109% higher than the cuttings of 10 cm in length (Figure 3). The hardwood cuttings of 20 and 30 cm in length did not differ in the number of leaves and primary roots; on average, they were around 88% and 101% higher than the cuttings of 10 cm in length, respectively (Figure 3). The hardwood cuttings of 20 cm in length had a root dry mass around 95% higher than the cuttings of 10 cm in length (Figure 3). No difference was observed between the cuttings of 10 and 30 cm in length for the root dry mass. The cuttings of 20 and 30 cm in length did not differ concerning the dry matter of shoots and the total, and, on average, were about 85% and 80% higher than the cuttings of 10 cm in length (Figure 3).

### 3.2. Micronutrients and Plant Regulators

There was no influence of the interaction between immersion of cuttings in solutions and parts of the branch where the cuttings were obtained on any of the variables at 77 days after planting. The solutions with the micronutrients or plant regulators influenced the survival rate, percentage of rooting, number of primary roots, and root dry matter. The parts of the branch influenced the survival rate and percentage of callused cuttings.

The lowest percentage of rooted and sprouted cuttings and the survival rate occurred with the immersion of the cuttings in the GA3 solution. The other treatments did not influence the percentage of rooted cuttings and survival rate. The highest percentage of sprouted cuttings occurred with the immersion in water and solutions with 2,4-D, biostimulant, and zinc. The immersion in water or solutions with micronutrients or plant regulators did not influence the percentage of callused cuttings (Figure 4).

The immersion of cuttings in the GA3 solution resulted in the lowest number of sprouts and leaves, about 80.5% and 56.6% lower than the average of the other treatments for the number of leaves and shoots, respectively. The highest number of primary roots was observed in the immersion in the 2,4-D solution. The lowest number of primary roots was found in the immersion in the GA3 solution (Figure 4).

The highest values of root dry matter were found in immersion in water and the solutions of 2,4-D, biostimulant, and zinc. The immersion of cuttings in water and solutions of micronutrients or plant regulators did not influence the dry matter of shoots and the total (Figure 4).

The highest survival rate and percentage of sprouted and callused cuttings were observed in the cuttings from the basal part of the branch. The percentage of rooted cuttings was not influenced by the part of the branch where the cuttings were extracted (Figure 5).

There was no influence of the interaction between the immersion of the cuttings in the solutions and the part of the branch where the cuttings were obtained on any of the variables at 120 days after planting. The part of the branch where the cuttings were extracted influenced the number of primary roots, shoots dry mass, roots dry mass, and total dry mass of *Jatropha curcas* L. saplings at 120 days after planting. The immersion solutions influenced all of the variables of *Jatropha curcas* L. saplings at 120 days after planting.

At 120 days after planting, the lowest number of sprouts, leaves, and primary roots occurred with the immersion of cuttings in the GA3 solution, about 77.8%, 86.3%, and 94.8% lower than the average of the other treatments, respectively (Figure 6).

The lowest values of the roots, shoots, and the total dry matter were found in the immersion of cuttings in the GA3 solution, about 81.8%, 76.4%, and 76.4% lower than the average of other treatments (Figure 6). The saplings produced with cuttings from the middle part of the branch had a number of sprouts 29.7% higher than the cuttings from the apex. The saplings produced with cuttings from the basal part had the number of primary roots 40% higher than the cuttings taken from the apex. There was no influence of the part of the branch on the number of leaves of the saplings at 120 days after planting (Figure 6). The saplings produced with cuttings from the basal part had the dry matter of roots, shoots, and the total dry matter of about 54.0%, 42.4%, and 45.2% higher than the cuttings taken from the apex, respectively (Figure 6).

## 4. Discussion

### 4.1. Hardwood Cutting Length

Vegetative propagation by cuttings has numerous advantages, one of which is to provide the implantation of more uniform and agronomically superior commercial plantations than those obtained by heterozygous seeds [26]. The plants propagated vegetatively by this method showed similarities in size, yield, and synchronicity of fruit maturation [22,26]. The plant organs, such as branches, and stems, when cut from the parent plant can develop roots in a short period. However, the quality and quantity of roots formed can vary depending on several factors (part of the cut on the branch/stem, tissue lignification, plant age, and others), therefore it is necessary to define the best technical procedures to increase the plant yield [28,29]. In this sense, the standardization of the cutting postion and cutting length was directly related to the energy reserve and plant hormones available in the cutting tissues [30,31]. Thus, the cutting length and location of the cutting in the plant directly influenced the rooting, survival, and quality of the saplings produced.

In the current study, the hardwood cuttings’ length significantly affected the survival rate, where the hardwood cuttings of 10 cm had the lowest survival rate. Similarly [32], verified that the survival rate was inferior in the cuttings of 10 and 15 cm in length, with a 59% survival rate for cuttings of 10 cm. The longer hardwood cuttings generated saplings with a higher number and/or dry mass of shoots and primary roots. These results are attributed to the greater number of buds and reserves accumulated in the longer hardwood cuttings. A study by [33], also indicated that longer cuttings had the highest rooting percentage and dry mass of roots. The similar results were also verified by [34], evaluating cuttings from 20 to 40 cm in length. These results are related to the greater amount of reserve [35,36]. It was also observed that the longer cuttings had a higher number of sprouts and leaves and a larger dry mass of shoots and roots. Consequently, they had a higher number of leaves, dry mass of shoots, and total dry mass. In addition, the longer cuttings have a greater amount of reserves to supply the demands of the physiological drains in the cuttings (buds and roots formation), this contributes to the greater survival rate of cuttings [24,31,34].

The hardwood cuttings at different parts of the branch did not influence the variables analyzed in this study. The accumulated reserve will be used in the process of rooting and sprouting. Due to the large accumulation of reserves in the hardwood cuttings, no differences were observed in the formation of the saplings by the cuttings from the different parts of the branch. In line with our findings, the part of the branch to produce cuttings does not influence the initial growth of the *Jatropha curcas* L. saplings. Similar results were observed between the hardwood cuttings of 20 and 30 cm in length. These small cuttings have less water loss and make the transport and handling of the propagation material easier [33]. Overall, for the vegetative propagation of *Jatropha curcas* L., it is suggested to use hardwood cuttings of 20 cm in length because it is possible to produce 50% more saplings with the same number of branches than the hardwood cuttings of 30 cm in length.

### 4.2. Micronutrients and Plant Regulators

According [32], one of the difficulties for propagating *Jatropha curcas* L. through cuttings is that only fine roots originate in the cuttings, which gives the plants reproduced by this method less wind resistance. However, it is known that several factors can influence the proper development of plants propagated through vegetative reproduction, such as carbohydrate reserves, plant hormones, and mineral nutrients [37]. In this sense, mineral nutrition is one of those crucial factors necessary for the healthy development of productive plants. A proper balance of micronutrients is necessary, especially with micronutrients capable of playing an active role in the rooting process because these micronutrients are necessary for the formation of plant tissues, including their enzymatic reactions: osmoregulation, photosynthesis, and carbohydrate metabolism [38,39]. Among these nutrients, boron and zinc are relevant as it is known that they can activate root formation in several plant species and could therefore contribute to the rooting of the *Jatropha* cuttings [24]. All of the processes described above are mediated and regulated through plant hormones, for instance the role of auxin in plant development that acidifies the cell walls, providing their weakening to facilitate the absorption of water and solutes and stimulating the growth process and the synthesis of polysaccharides for the formation of new cell walls [31]. The root development is influenced by growth regulatory substances, with auxins being the primary regulator responsible for elevating the formation of the root primordia [25,40]. However, the *Jatropha curcas* L. plant has an indeterminate growth pattern, the occurrence of the apical dominance provided by auxin may inhibit the development and formation of lateral buds. In turn, in some plant species, gibberellin (GA3) can promote the overcoming of apical dominance [41]. Therefore, we hypothesized that applying low concentrations of gibberellin could delay the development of apical buds, favoring the development of lateral buds and roots by reducing the consumption of energy reserves by the apical buds.

In this experiment, the boron treatment did not differ from immersion in water for the evaluated characteristics at 120 days. There are studies with positive results from using the mixtures of boron and auxin in the rooting of cuttings [42,43,44].

In general, the treatment with immersion in a solution with zinc did not differ from that in water, corroborating the work of [45]. They did not verify the influence of zinc on the rooting and sprouting of cuttings of *Platanus acerifolia* Ait. These results may have occurred due to the absence of auxin application together with zinc, as indicated in the study of [46], who concluded that zinc favors the rooting of cuttings. Zinc is essential for the synthesis of tryptophan, a precursor of indoleacetic acid (IAA) responsible for root formation [47].

There was no emission of primary roots in the cuttings treated with immersion in GA3 solution; consequently, there was a lower percentage of sprouted cuttings. Even though there were shoots in the cuttings, they did not have roots to absorb water and keep the hardwood cuttings alive, resulting in a lower percentage of live cuttings. In this regard, gibberellin (GA) inhibits the rooting process of cuttings [26,48,49,50,51,52] or in some cases slows down this process [53]. Moreover, gibberellin affects the transcription processes [26] and inhibits cellular dedifferentiation [54]. There is an inverse relationship between the content of gibberellins in cuttings and the capacity for root formation [52]. Some studies show positive results of rooting in cuttings with paclobutrazol that inhibits gibberellin synthesis, such as [52,55]. The gibberellic acid can stimulate root formation in specific cases [56,57]. However, it depends on the stage of root development and the environmental conditions to which the cuttings are subjected [26,31]. The cuttings immersed in the solution with 2,4-D, a synthetic auxin, had the highest number of primary roots at 77 days after planting and accelerated the emission of primary roots. However, after 120 days, except for the solution with GA3, the number of primary roots was similar among the treatments. Auxin is synthesized at the apex of the stem and in younger leaves and is then transported to sites of action in the plant [58], and plays a key role in the root formation [40,59,60,61]. When auxin is applied, endogenous auxin concentration in the cutting increases continuously until the rooting process begins [62,63]. The cuttings immersed in 2,4-D solution received auxin directly at the site of root formation, resulting in the highest number of primary roots. In parallel, according to [64], the exogenous application of auxin promotes a more efficient mobilization of endogenous auxin. Auxin also influences the basal carbohydrate accumulation, which is directly related to the rooting process [65]. Some studies with *Jatropha curcas* L. indicate that the use of auxin in the form of indoleacetic acid (IAA), indole-3-butyric acid (IBA), and 1-naphthaleneacetic acid (NAA) increased the number of rooted cuttings and the number of roots per cuttings [64,66,67] as found in this study. The commercial product Stimulate^®^ has an auxin/cytokine ratio of 1:1.8. According to the study of [68], this relationship is considered intermediate. Thus, it does not favor the formation of roots and shoots, with only the appearance of undifferentiated cells. The interaction between auxin and cytokinin plays a vital role in controlling some of the developmental processes, such as the formation and maintenance of meristems [69,70]. A high auxin/cytokinin ratio promotes root system growth, while a low auxin/cytokinin ratio favors shoot development [68]. The intermediate auxin/cytokinin ratio favors the development of undifferentiated tissues, known as calluses [68].

### 4.3. Effect of Nutrients and Hormonal Treatment Was the Part-Specific

The part of the branch influenced the number of primary roots and sprouts and the dry matter of roots, shoots, and total weight. The basal and middle parts showed the greater reserve present in these cuttings, which have a larger diameter than the cuttings from the apex (Figure 3). The amount of carbohydrates and substances that promote or inhibit the formation of roots and sprouts ranged along the branch length [24,26,31,34]. In this way, boron, and zinc may have interacted with the carbohydrates and hormones in the cuttings, resulting in different behaviors for each part of the branch. If the volume is considered, the cuttings from the basal part have a volume around 63% greater than the cuttings from the apex. The volume of the cuttings is directly related to the reserve accumulated in them. The greater amount of reserve in the basal cuttings originated saplings with a higher vigor than the cuttings from the apex. In a study conducted by [33], it was found that the cuttings from the basal part that originated from the *Jatropha curcas* L. plants with a greater number of sprouts and branches had a greater dry mass of roots and shoots.

In conclusion, some of the points should be highlighted based on the results of the two experiments. The use of 2,4-D is an important tool to obtain a high survival rate of hardwood cuttings, since it accelerates the emission of primary roots. In addition, 2,4-D has the advantages of being easy to obtain and of low cost. Even with the highest survival rate, the use of cuttings from the apex of the branch should be avoided when possible, since the quality of the saplings produced is lower than the cuttings of the basal part. The survival rate is an important aspect; however, the quality of the saplings will directly influence the success of the plantation after the transplanting of saplings. Regarding the cutting length, we consider the use of cuttings of 20 cm in length the best option since their results were better than the cuttings of 10 cm and equal to cuttings of 30 cm in length. Considering only the cuttings of 20 and 30 cm in length, even with similar results, the use of hardwood cuttings of 20 cm in length is more advantageous since it will result in 50% more saplings with the same number of branches than the hardwood cuttings of 30 cm in length.

## 5. Conclusions

The *Jatropha curcas* L. plant has great potential to be used as a model plant in several studies involving native forest species. Its rapid growth, precocity, and adaptability facilitate silvicultural studies, allowing the obtaining of important information in a short time, and reducing labor costs. Immersion in a low concentration of 2,4-D solution stimulates the emission of primary roots of *Jatropha curcas* L. using hardwood cuttings. For the vegetative propagation, it is recommended to use cuttings of 20 cm in length extracted from the basal part of the branch with immersion in 2,4-D solution. The boron and zinc treatment did not differ from immersion in water for the evaluated characteristics at 120 days. The immersion of the *Jatropha curcas* L. cuttings in the GA3 solution is not recommended.

## Figures and Tables

**Figure 1 plants-11-02457-f001:**
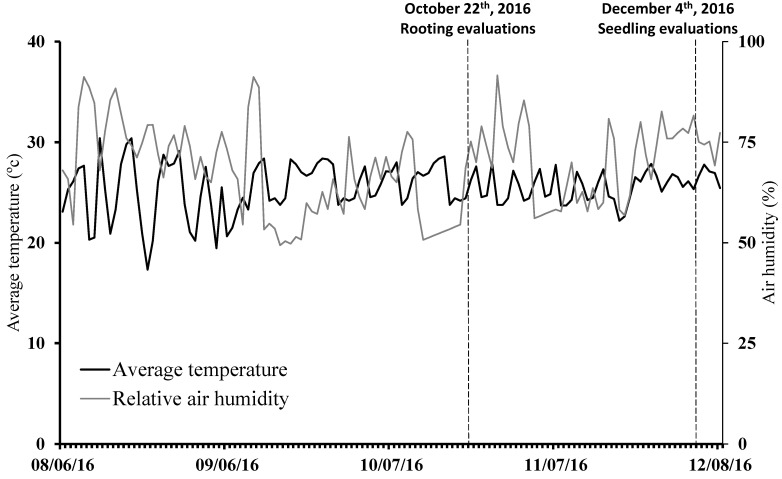
Average temperature and relative air humidity inside the agricultural screenhouse during the experiment.

**Figure 2 plants-11-02457-f002:**
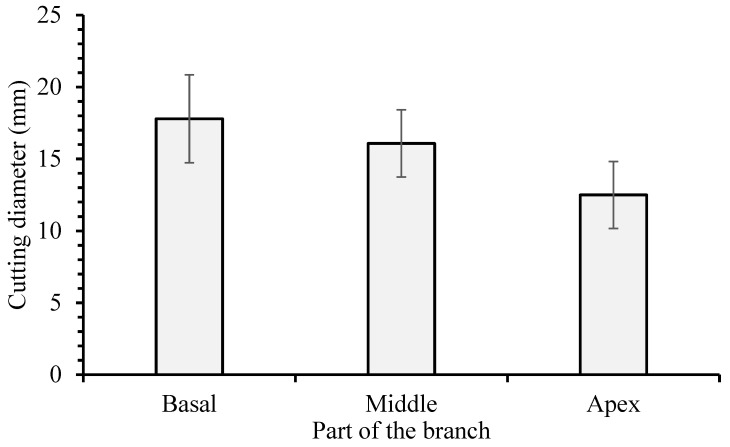
The diameter of *Jatropha curcas* L. hardwood cuttings at different parts (basal, middle, and apex) of the branch, collected in the upper third of the plant. Values are represented by the mean ± standard deviation of at least three independent replicates.

**Figure 3 plants-11-02457-f003:**
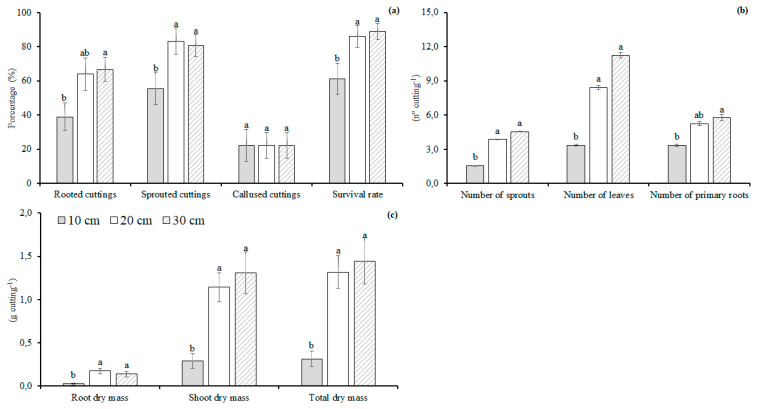
Percentage of rooted, sprouted, and callused cuttings, survival rate (**a**); number of sprouts, leaves, and primary roots (**b**); roots, shoots, and total dry matter (**c**) of *Jatropha curcas* L. hardwood cuttings with different lengths at 77 days after planting. Different letters above each bar indicate significant differences among the hardwood cuttings with different lengths by the Least Significant Difference (LSD) test at the 5% probability level. Values are represented by the mean of four independent replicates.

**Figure 4 plants-11-02457-f004:**
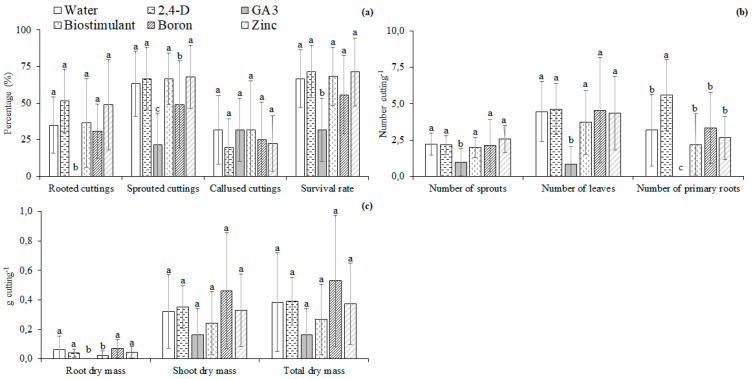
Percentage of rooted, sprouted, and callused cuttings, survival rate (**a**); number of sprouts, leaves, and primary roots (**b**); roots, shoots, and total dry matter (**c**) of *Jatropha curcas* L. hardwood cuttings immersed in distilled water or solutions of micronutrients or plant regulators at 77 days after planting. Different letters above each bar indicate significant differences among the immersion in water or solutions of micronutrients or plant regulators by the Least Significant Difference (LSD) test at the 5% probability level. Values are represented by the mean of four independent replicates.

**Figure 5 plants-11-02457-f005:**
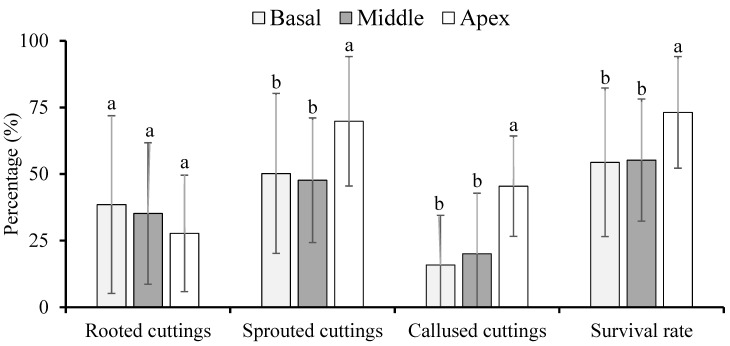
Percentage of rooted, sprouted, and callused cuttings and survival rate of *Jatropha curcas* L. hardwood cuttings extracted from different parts of the branch at 77 days after planting. Different letters above each bar indicate significant differences among the parts of the branch where the hardwood cuttings were extracted by the Least Significant Difference (LSD) test at the 5% probability level. Values are represented by the mean of four independent replicates.

**Figure 6 plants-11-02457-f006:**
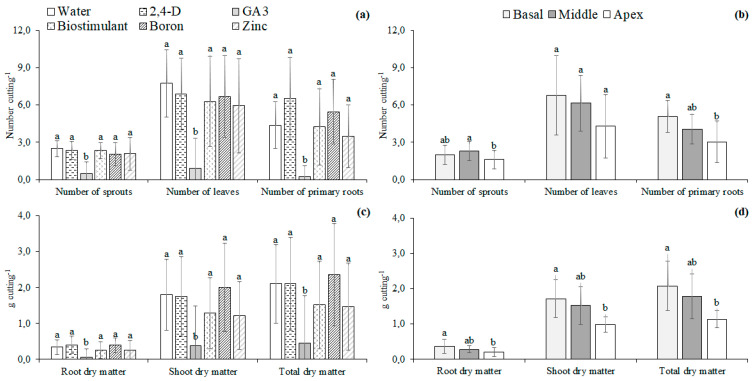
Number of sprouts, leaves, and roots (**a**,**b**); roots, shoots, and total dry matter (**c**,**d**) of *Jatropha curcas* L. saplings produced from hardwood cuttings immersed in distilled water or solutions of micronutrients or plant regulators and extracted from different parts of the branch at 120 days after planting. Different letters above each bar indicate significant differences among the immersion in water or solutions of micronutrients or plant regulators and among the parts of the branch where the hardwood cuttings were extracted by the Least Significant Difference (LSD) test at the 5% probability level. Values are represented by the mean of four independent replicates.

## Data Availability

The datasets used and/or analyzed during the current study are available from the corresponding author on reasonable request.

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
