# Peer review of "Jatropha curcas L. as a Plant Model for Studies on Vegetative Propagation of Native Forest Plants"

_plants, 2022, doi:10.3390/plants11192457_

Round 1

Reviewer 1 Report

The authors have investigated factors that influence the success of propagation of Jatropha curcas by hardwood stem cuttings. This is an important species for a number of reasons, so improving propagation protocols is a reasonable mission. I am not personally familiar with the species or its ease of propagation. The authors investigated cutting length, position, and nutrient and hormone concentrations. There are some interesting results here, and the work provides some specific recommendations for propagators.  

Below are specific comments to improve the manuscript:

The title is very long and somewhat confusing. Is this about a model species, or simply about experiments to propagate Jatropha curcas? In any case, I would streamline to less than half its current length if possible.

Introduction:

The selection of Jatropha as a model plant in the abstract and introduction are perplexing.  Is it already considered a model plant, or are the authors proposing it as a model plant?  If this is central to the manuscript, I’d like the authors to specifically describe how its characteristics make it a representative model for a broad family of plants. I’m unsure whether the authors are proposing it as a model species for studies in propagation, or as a model species for other purposes (and propagation becomes important for this reason)?Overall, the model species aspect of the manuscript only confuses the narrative of the paper and ought to be removed in its entirety. At its core, this is a manuscript about propagation of one species, and it should be presented as such.  

Most of lines 41-53 in the abstract should be removed

In general, much of the language in the introduction section is verbose, and could be substantially reduced. For example, the first 4 or 5 paragraphs (lines 74-105) could be reduced by 50-70% into one or two streamlined paragraphs with fewer citations.

Lines 106-112 - this paragraph is out of place and disconnected from the rest of the narrative.

Line 125: What, specifically, about its fast growth and reproduction makes Jatropha a good model plant for reproduction of native plants in general? I caution the authors against (somewhat arbitrarily) selecting Jatropha as a general purpose model plant. Doesn’t a model plant become a model plant when pivotal studies establish it as a model plant in a specific context? Or is Jatropha already a model plant?  

For a paper on propagation, the introduction is quite long.  

Lines 138-142: paragraph could be reduced to one sentence.  

Lines 148-153: As written, the paragraph implies that directional transport of hormones is the only, or primary, reason that different portions of branches root with different degrees of success. We know carbohydrates and other factors play a role, something the authors state in the subsequent paragraph. I’d combine the two paragraphs, streamline them, and only list hormone concentrations among the several factors that may influence rooting success of cuttings of different types.

Line 156: hardwood cuttings are more lignified, but I’m not sure that this generally means a more difficult rooting process.  Slower, maybe, but not always more difficult. And for hardwood cuttings, the slower process might be related to factors other than lignification (e.g., degree of biological activity, hormone concentrations, etc.)

Line 158: I’m still unclear whether it would be a model plant for propagation studies of forest plants, or whether successful propagation would establish it as a model plant for other purposes?

I would present the statement of objectives more explicitly and clearly.  For example, “Our objectives were to…”

Materials and Methods:

Line 170 and elsewhere in the manuscript - authors refer to plants produced by cutting propagation as “seedlings”.  This is categorically incorrect for plants that were not propagated by seed. 

Line 172: both experiments were conducted in duplicate, so there are 4 experiments conducted in total? 

Line 174: seedling production, i.e. the cutting-propagated plant produced seeds, which then produced new seedlings?

Line 184: When the factorial experiment is indicated, the factors should be clearly identified at that point.  

Line 184: The authors write that there were four replicates for Experiment 1 and four replicates for Experiment 2. However, what constitutes a replicate is not clear. One cutting? 10 cuttings? All the cuttings from an individual plant? If there is more than one cutting per replicate, then the experiment has subsamples.  If they exist, are subsamples analyzed correctly? If there are no subsamples, then are results based on a sample size of four cuttings per treatment combination?

Line 185: difficult for the reader to follow this without more detail. How many stem cuttings from each portion of the branch? Are all cuttings from basal portion 10 cm, all middle cuttings 20 cm, and all apex (terminal) cuttings 30 cm? Or are different cuttings from each position of the stem 10, 20, and 30 cm?  

How many plants served as the source of stem cuttings? In what condition were those plants growing? Where did they originate?

Line 199 - the 6 x 3 factorial should be spelled out more clearly, e.g., “immersion for four hours in water or one of five treatment solutions”

Line 200 - I didn’t see where the authors indicated how the cuttings were immersed. The entire cutting was submerged, or the basal ends of cuttings were submerged?

Lines 204-215 - it is hard at this point in the narrative to determine whether micronutrients and hormones were investigated in different combinations vs. only individually.

Lines 187-193 and Lines 215-222 - The same paragraph is repeated verbatim, as far as I can see.  The materials and methods should be restructured to eliminate this redundancy.  

Line 227 - was the soil/poultry litter/sand substrate used to root cuttings, or just to study seedling growth? The latter is implied by this sentence.  

The materials and methods section does not describe how cuttings were managed with respect to irrigation. This would have a profound effect on rooting in general, and on the success of different substrates. Did the plastic bags have holes? Were cuttings inserted in bundles or one cutting per bag? Does the bag with cutting(s) represent a replication?

How was analysis of variance conducted? Was the experiment analyzed as a completely random design or as a randomized complete block design? What software was used? 

The randomized complete block design has one replicate of each treatment combination within each block. Again, what is a replicate here?

Results: 

The authors report that cutting length impacted rooting results. In the materials and methods, cutting length and position within the stock plant’s shoot seemed to be two separate factors, but here only cutting length is mentioned. Were cuttings from different parts of the shoot pooled for analysis, or did I misunderstand something here? If only cutting length was considered (and not length + portion of the branch), what is the other factor in the 3x3 factorial?

Figure axis labels should be larger.  Lettering is small and hard to read.  

Lines 252-253: the topic sentence of the paragraph is about rooting, and then the next sentence is about sprouting.  Is sprouting in reference to roots or shoots? Normally, we tend to think of sprouting as something that happens to shoots.  I would revise language to be more precise.  

Line 281 - “there was no influence of the interaction” — this is the same as writing there was “no interaction”?

Figures - I would like to see the number of individual cuttings included in the figure captions so the reader understands how the means were obtained.  

Discussion:

The discussion is structured in an unusual way. Most discussions begin with a paragraph that highlights the conclusions (main findings) of the current study, and then elaborates on the specific results to put them into the context of the existing literature. Here, the first several paragraphs read to me as introduction material.   

First paragraph (lines 346-350) can probably be deleted

Second and third paragraphs (351-364) can be substantially shortened and streamlined

Line 368: what was similar? 59% rooting specifically, or reduced rooting compared with longer cuttings? Same species? What lengths were Severino et al. (2011) comparing?  

In the paragraph from 383-395, the authors revisit the idea of cuttings from different parts of the branch for Experiment I. This factor was missing from the results section for Experiment I, as far as I could see.

Lima et al., 2010 studied seedlings or cuttings?

Lines 397-425 — all of this reads as introduction content. 

Section 4.3 - the first paragraph beginning on line 489 can be reduced by >70%

My overall assessment of the discussion section is that the narrative is very long and a little hard to follow.  

Conclusions:

Model plant sentence is unrelated to the experiments in the manuscript. 

Literature Cited:

63 references is a lot of references for a paper about propagation by stem cuttings!  

Author Response

Thank you very much for all the comments, they helped a lot to improve our work.

Reviewer 2 Report

General comments

Manuscript seems to me interesting based on the use of Jatropha curcas as a plant model for studies on vegetative propagation. Authors should consider all comments and recommendations in order to improve the manuscript.

The English language needs improvement. I suggest a native English language person should review the document.

Title and Keywords. I think the Authors should reduce the title because is too long (38 words). I suggest “Jatropha curcas, as a plant model for studies on vegetative propagation of native forest plants” as a final title. Other words from the original title can be used as keywords. I suggest eliminate “model plant”, “native species” and “vegetative propagation” from the keywords since they are already included in the Title.

Highlights. Second point should be: “The immersion in 2,4-D solution accelerated the emission of primary roots in hardwood cuttings”.

Abstract. I consider that this section should be reduced. It is too long (30 lines, 459 words). I suggest to use no more than 300 words. The most important part for an abstract is Results section. Much of the information from the first 14 lines can be eliminated and reduced to only 3 lines for this introductory section.

Introduction. This section can include some of the information eliminated from the introductory part of the abstract. At the end of this section, I consider that the objective of the study is not clear since the evaluation of some variables is a mean to reach the objective. Last paragraph of this section can be reduced to make it more clear. For example, you can group some variables as “propagation techniques”.

Methods. In this section there is lack of important information. Authors should consider include more detailed information in the methods section to clarify production conditions. i.e., in the hardwood cutting length experiment, substrate used, containers characteristics, irrigation, experimental unit?, etc., in order to improve understanding of the results.

Page 6, line 221, again, how many cuttings were used? And experimental unit.

Results.

Page 7, line 252, eliminate “(30 cm)”, since 20 and 30 cm cuttings were statistically similar treatments.

Variation levels of the treatments showed in Figure 4 is probably related with a small experimental unit (probably only four cuttings as I can understand). That is the reason this information is important in the methods section.

Discussion. I consider this section enough developed. However, I suggest include more recent cites.

References:

There is only 8 references (articles) from last 5 years. I suggest do an exploration more in deep and include recent published information.

There are many cites not included in the references section. Some examples are:

RECH et al., 2018 is not included in the references section.

ALFENAS et al., 2009 is not included in the references section.

DIAS et al., 2012b should be included as “DIAS et al., 2012” in all document.

BARBEDO et al., 2018 is not included in the references section.

FERRARI et al., 2004 is not included in the references section.

SIMAO et al., 2007 is not included in the references section.

Correct Authors and format ot this cite: Dranski, j. A. L., pinto-júnior, a.s., steiner, f., zoz, t.., malavasi, u. C., malavasi, m., guimarães, v.f. PHYSIOLOGICAL MA- 584  TURITY OF SEEDS AND COLORIMETRY OF FRUITS OF JATROPHA CURCAS. Revista Brasileira de Sementes, Londrina,  585  v.32, n.4, p. 158- 165, 2010.

NEVES et al., 2006 is not included in the references section.

Authors should review in detail and be sure that all publications mentioned in the document are correctly cited in the references section.

Author Response

(The authors gave the same response as above.)

Round 2

Reviewer 1 Report

I appreciate that the authors made an effort to improve the manuscript following the first review. I especially appreciate the additional experimental details and statistical description. The most significant remaining challenges are the brevity and clarity of communication. Many facts cited in the introduction and discussion can be eliminated. The use of model species to frame the project still doesn't make sense to me for a study strictly focused on some hardwood propagation experiments with this plant. The introduction is verbose and the discussion section is probably twice as long as is appropriate. 

The revised title implies to me that the paper will be a review of what is currently known about the use of Jatropha curcas as a model species for vegetative propagation. But it’s really a paper reporting on two experiments to propagate Jatropha curcas from hardwood cuttings. I would not use this title. Something as simple as "Propagation of Jatropha curcas by hardwood stem cuttings..." seems more appropriate.   

It remains unclear - is Jatropha studied here as 1) a model plant for general future experiments with forest species (i.e., the ability to propagate it would allow its use as a model species in forestry, or 2) a model species for propagation of forest species (i.e., knowing how to propagate it somehow informs propagation of other plant species)? The objectives statement of the introduction and the conclusions statement imply the former, but title and methods and results imply the latter. I’m not sure that I would include any discussion of model species at all for a straightforward propagation manuscript. What motivates this rhetorical decision? In addition, the conclusions statement makes conclusions about its suitability as a model species due to factors other than those studied in the manuscript; how did these conclusions arise?  

To be published, the writing needs extensive revision by an editor for clarity and brevity, and standard use of language and punctuation. 

One crucial requirement of this revision was that it should be shorter than the very long first draft. Instead, this one seems to be even longer.

Author Response

We thank you for your contributions and hope that the justification we present in the attached document can clarify the remaining doubts.
